# The Autumn Low Milk Yield Syndrome in High Genetic Merit Dairy Cattle: The Possible Role of a Dysregulated Innate Immune Response

**DOI:** 10.3390/ani11020388

**Published:** 2021-02-03

**Authors:** Massimo Amadori, Chiara Spelta

**Affiliations:** 1RNIV, Italian Society of Veterinary Immunology, 25125 Brescia, Italy; 2Private Veterinary Practitioner, 27100 Pavia, Italy; chspelta@icloud.com

**Keywords:** dairy cattle, milk production, autumn, innate immune system, metabolic stress

## Abstract

**Simple Summary:**

Milk yield worldwide is dominated by few cosmopolitan dairy cattle breeds producing high production levels in the framework of hygiene standards that have dramatically improved over the years. Yet, there is evidence that such achievements have gone along with substantial animal health and welfare problems for many years, exemplified by reduced life expectancy and high herd replacement rates. Also, these animals are very susceptible to diverse environmental stressors, among which hot summer climate plays a central role in the occurrence of diverse disease cases underlying early cull from the herd. Milk production is also affected by heat stress, both directly and indirectly, as shown by low milk yield in the following autumn period. This article highlights the low milk yield syndrome and sets it into a conceptual framework, based on the crucial role of the innate immune system in the response to non-infectious stressors and in adaptation physiology at large.

**Abstract:**

The analysis of milk yield data shows that high genetic merit dairy cows do not express their full production potential in autumn. Therefore, we focused on metabolic stress and inflammatory response in the dry and peripartum periods as possible causes thereof. It was our understanding that some cows could not cope with the stress imposed by their physiological and productive status by means of adequate adaptation strategies. Accordingly, this study highlights the noxious factors with a potential to affect cows in the above transition period: hot summer climate, adverse genetic traits, poor coping with unfavorable environmental conditions, outright production diseases and consequences thereof. In particular, the detrimental effects in the dry period of overcrowding, photoperiod change and heat stress on mammary gland development and milk production are highlighted in the context of the autumn low milk yield syndrome. The latter could be largely accounted for by a “memory” effect on the innate immune system induced in summer by diverse stressors after dry-off, according to strong circumstantial and indirect experimental evidence. The “memory” effect is based on distinct epigenetic changes of innate immunity genes, as already shown in cases of bovine mastitis. Following a primary stimulation, the innate immune system would be able to achieve a state known as “trained immunity”, a sort of “education” which modifies the response to the same or similar stressors upon a subsequent exposure. In our scenario, the “education” of the innate immune system would induce a major shift in the metabolism of inflammatory cells following their reprogramming. This would entail a higher basal consumption of glucose, in competition with the need for the synthesis of milk. Also, there is strong evidence that the inflammatory response generated in the dry period leads to a notable reduction of dry matter intake after calving, and to a reduced efficiency of oxidative phosphorylation in mitochondria. On the whole, an effective control of the stressors in the dry period is badly needed for better disease control and optimal production levels in dairy cattle.

## 1. Introduction

Dairy farming activities are presently based on few cosmopolitan cattle breeds [1] with a high genetic improvement for milk yield. In the presence of large increases in milk yield and substantial improvements of milk production hygiene over some decades [2,3], the health and welfare status of dairy cows still raises serious concerns because of low fertility and life expectancy [4]. In addition to that, there has been a dramatic improvement in milk quality (lower bulk milk cell counts), which has not coincided with a reduced prevalence of disease cases [2] and, in particular, of clinically-overt mastitis [3]. This still causes large antibiotic usage in dairy farms as both therapeutic and prophylactic treatments at dry-off. These practices have been of concern for many years, following human disease cases sustained by drug-resistant bacterial strains isolated from cattle [5].

On the whole, strong circumstantial and experimental evidence shows that farm hygiene and technological improvements of primary production have not overcome fundamental risk factors for disease and early cull from the herds, mainly associated with mastitis, metritis, placenta retention, abomasal displacement, ruminal disorders. These are often collectively referred to as “production diseases” since their prevalence is strictly linked to high milk yields. Accordingly, the average number of calvings of Holstein Frisian cattle is often <2.5, which corresponds to a life expectancy <5 years [6]. In addition to the ethical concerns associated with the above conditions [7], farmers often sustain high direct and indirect losses, as well as high social costs; these may question the very sustainability of the present dairy farming activities [8].

On the whole, the above findings point at metabolic stress of dairy cows as the crucial element underlying occurrence and prevalence of production diseases. Metabolic stress can be defined as a disequilibrium in the homeostasis of a living organism consequent to an anomalous utilization of nutrients [9]. As for high-yielding dairy cattle, disequilibrium stems from the imbalance between energy needed for lactation and energy provided through dry matter intake, which causes a substantial and prolonged Negative Energy Balance (NEB) [10]. As explained in a recent review paper [1], the innate immune system is pivotal to sensing metabolic stress. Accordingly, it can sense changes of cellular energy status (Adenosine triphosphate, ATP/Adenosine monophosphate, AMP ratio), shortage of amino acids, reduced oxygen tension, oxidative stress, as well as high concentrations of metabolites like non-esterified fatty acids (NEFA). The impact of these stressors is definitely amplified following the release of Damage-associated Molecular Patterns (DAMPs) from stressed cells, which signal to the innate immune system a danger condition [11]. Therefore, the inflammatory response of dairy cows around calving [12] can be set in a robust conceptual framework, in which the innate immune response to non-infectious stressors [13] plays a fundamental role.

In this conceptual framework, the type and height of the response will be affected by the previous interaction of the host with the same or similar stressors, in what might be defined as innate immune memory [14]. In practice, the innate immune system keeps the “memory” of previous stressors and modifies accordingly the profile of its response following a new exposure. As for dairy cows, the inflammatory response to a plethora of stressors during the dry period predisposes to an enhanced inflammatory response after calving and higher disease prevalence [15]; also, the interferon (IFN)-gamma response to common microbial agents like *Mycobacterium avium* subs. paratuberculosis in the dry period can be conducive to ketosis and related metabolic disturbances after calving [16].

It has been clearly demonstrated that higher inflammatory responses after calving are conducive to lower milk yields [15]; this is probably due to a substantial worsening of energy efficiency (10–15%) under inflammatory conditions [17]; this is in agreement with mitochondrial dysfunctions during severe NEB observed by a bovine oligonucleotide array [18]; also, oxidative stress and inflammation complement and potentiate each other by previously defined molecular mechanisms [19]. Reduced milk yield of dairy farms in autumn has been described for a long time [20]; more recently, this was characterized as “Autumn low milk yield syndrome” by Tondo and Fantini in Italy (see https://www.ruminantia.it/wp-content/uploads/2016/05/LA-SINDROME-DELLA-BASSA-PRODUZIONE-DI-LATTE-IN-AUTUNNO.pdf), which is fully relevant in our opinion to the above conceptual framework. This syndrome is characterized by the inability of dairy cows to express their full production potential in autumn. This way, the same herd shows reduced milk yield in autumn compared to spring, in the presence of the same average number of lactation days, thermoneutral temperatures, parity level and feeding profile.

On the whole, the authors believe that the low milk yield syndrome could be accounted for by the aforementioned “memory” effect of the stressors experienced in the summer period, adding to the metabolic stress of the early lactation period. In this respect, the paper presents a hypothesis and the relevant supporting elements, but no specific experimental study or meta-analysis testing the hypothesis itself. Accordingly, the authors will parse the scientific findings underlying this hypothesis as well as the circumstantial evidence arising from field remedies and empirical approaches. The authors will also highlight a few points of possible weakness of this hypothesis and indicate future paths of fundamental and applied research to provide conclusive answers to pending issues. Some translational aspects related to dairy farming activities will be dealt with as well.

## 2. Characterization of the Autumn Syndrome

This syndrome was defined on the basis of the official controls carried out by Associazione Italiana Allevatori (the Italian Farmers Association or AIA) on lactating Friesian cattle. These reports consistently show that the average number of Days in Milk (DIM) is low in Friesian cows in the spring period, since a large number of animals have just calved, and milk production is high. On the contrary, in summer, average DIM tends to increase, and production decreases accordingly. Also, this decrease seems to be justified by high temperatures, lower ingestion and less efficient management due to seasonal agricultural commitments in the countryside. Peak calving is observed in late summer to early autumn as a consequence of the summer anestrus [21] in the previous year, which causes plenty of cows to get pregnant in autumn.

Until the whole summer period, milk production is mainly affected by the average DIM level and, therefore, by the distribution of calvings. This is not true of the autumn period. Despite the reduced average DIM (comparable to the spring levels), milk production remains low, without the expected increase. The average DIM in October and September are substantially equal to those of April and May, respectively. Yet, cows produce 2.5 to 2.7 kg of milk less per day per cow. This means that in a dairy cattle farm of 100 cows in lactation for a period of 90 days, about 22.5 tonnes of milk are lost (Figure 1). This trend occurs in a similar way every year, and we can find the same trend not only in Italy, but also in Europe and USA, and in a specular way in the southern hemisphere (Figure 2) (see https://www.ruminantia.it/wp-content/uploads/2016/05/LA-SINDROME-DELLA-BASSA-PRODUZIONE-DI-LATTE-IN-AUTUNNO.pdf). Please notice that feeding regimes of dairy cattle are based in Italy on Total Mixed Ration (TMR), with little if any access to pastures and spring grass. Most important, total energy and fiber to dry mass ratio of TMR show no difference between spring and autumn. As for the Austral hemisphere, the majority of dairy cows in New Zealand calve in a concentrated pattern in late winter-early spring (July to October) [22], which may be relevant to the different time-course of milk production with respect to Australia after the October peak (Figure 2). In this respect, the Australian time-course is reminiscent of the specular one observed in Europe and USA, whereas the time course of New Zealand is mainly accounted for by the seasonal calving pattern. Yet, a note of caution should be expressed about the Australian data set, as precise DIM normalization is lacking.

According to the same AIA data about the percentage of cows with production over 40 kg/day, it is evident that in the period from September to November, compared to spring, and under similar risk factor expression, the percentage of cows producing more than 40 kg/day of milk at the peak of lactation decreases.

Considering that cows at the peak of production between September and November are the ones that experience dry-off in summer, we examined the seasonal factors affecting the potential autumn production of these animals. Hence, the main stressors affecting dairy cows and the peculiarities of the summer period are described in the following sections.

## 3. Stressors Acting on Dairy Cattle

The non-lactating period and the subsequent peripartum are certainly most critical for dairy cows: they experience nutritional, metabolic, hormonal and immunological changes that have an impact on the prevalence of infectious and metabolic diseases [23].

### 3.1. Metabolic Stress

As recalled in the Introduction section, the nutrition of high genetic merit dairy cattle in the transition phase is characterized by demands that exceed the potential dietary intake [24]; this results in a temporary NEB, during which blood glucose decreases and body reserves are mobilized to provide energy; this results in an increase of NEFA in plasma. As long as this homeostatic adjustment remains efficient and balanced, fat mobilization is regulated. When this regulation is deficient, the cow will go into a state of metabolic stress with excessive concentration of plasma NEFA and overproduction of ketone bodies such as β-hydroxybutyrate (BHB). These conditions of metabolic stress are also associated with a dysfunction of the immune defenses, which makes the cow more susceptible to infectious diseases.

It has been observed, for example, that BHB has a harmful action on the antimicrobial mechanisms of leukocytes [25,26], induces oxidative stress, activates the cascade of proinflammatory signals with consequent damage to hepatocytes. Interestingly, NEB does not cause the same harmful effects in more advanced lactation stages, which point at complementation of diverse stressors in the peripartum period [1].

As described in humans, also cattle show the intersection of metabolism, low-grade inflammation and immune function [9]. Immune cells are directly involved in numerous metabolic functions [27], including the maintenance of gastrointestinal function [28,29], control of lipolysis of adipose tissue [30] and regulation of insulin sensitivity in multiple tissues [31]. This implies that the innate immune system can sense metabolic stress as a major threat to homeostasis [11] and acts to counteract its noxious consequences.

Resident immune cells are found in all tissues of metabolic importance and have a significant impact on the nutrient flow in adipose tissue. The cross-talk between adipocytes and immune cells [32] has a huge influence on the inflammatory status of the whole body and can provide a homeostatic regulation of immune response versus tolerance. Also, the inflammatory status can have surprisingly large impacts on milk yield [33].

### 3.2. Oxidative Stress

Under metabolic stress conditions high genetic merit dairy cattle also suffer from oxidative stress, i.e., accumulation of reactive oxygen metabolites (ROMs) with reduced antioxidant defences, resulting in damage to tissues and cells involved in the inflammatory response [34,35].

The balance between the inflammatory response and the mechanisms of its resolution is also lost, so that in the transition period (±3 weeks around calving) cattle may be in a systemic inflammatory state even without apparent clinical manifestations [12]. Oxidative stress and inflammation enhance each other [19], and act synergistically on precalving chronic inflammatory status. In order to re-establish an adequate antioxidant response and to activate the immune system, the cow must divert energy that could otherwise be used for milk production. Therefore, subjects with high inflammatory response have a lower energy conversion efficiency [15].

## 4. Inflammatory State and Thermal Stress

Having defined the negative events that can occur in animals in the peripartum period, we analyze hereunder the factors underlying poor production performance of cows dried in the summer period.

First, the inflammatory state in transition dairy cows is increased by thermal stress [36]. Please notice that the high merit cow, as a result of her metabolism, generates a considerable amount of heat. Therefore, these animals cannot adapt to the hot, humid summer climate, as shown e.g., by the dramatic impact of hot summers [37].

Exposure to high temperatures and humidity is conducive to disease occurrence (immunodepression, susceptibility to laminitis, mastitis, metritis, uterine prolapse), metabolic dysfunctions (acidosis, abomasum dislocation) and effects on production (decrease in milk quantity and quality). The negative effects of heat stress may persist for long periods despite the return to more favorable climatic conditions and project their negative effects to the next lactation [38], as well as on the productive and health performance of the daughters [39]. On the whole, heat stress displays effector mechanisms in the short term, but it also tends to perpetuate its effects over a long time, well beyond the actual time frame of the stressing condition.

### 4.1. High Temperature Effects on the Immune System

The results by Lacetera et al. [40] indicated that extremely high temperatures during the summer period (heat waves) may also be responsible for a profound shift from cellular to humoral immune responses, which affects disease resistance. On the other hand, previous studies have shown that the prevalence of certain infections in cattle is higher during the hot summer months. Accordingly, non-lactating, pregnant cows cooled in the same period show greater immunocompetence compared with uncooled subjects, as shown e.g., by enhanced lymphocyte proliferation in vitro [41].

### 4.2. Heat Stress and Changes in Liver Function

In their study, Skibel et al. [42] investigated the liver protein expression profile of postpartum cows that were cooled or heat-stressed during the dry period to gain insight into how protein expression is altered by prior heat stress and may contribute to production performance and disease outcomes. It was observed that heat-stressed cattle have impaired mitochondrial function and altered lipid, carbohydrate, and amino acid metabolism in the liver. It appears that cows subjected to heat stress have reduced ATP synthesis, greater oxidative stress, shifts in precursor supply for gluconeogenesis, and accumulation of lipids in their liver. Therefore, these changes in liver function may be associated with disease occurrence in the transition period and poor lactation performance.

## 5. Photoperiod and Heat Stress Actions on Mammary Gland Development

Long-day photoperiod and heat stress, characteristics of the summer season, seem to have synergistic actions in the dry period, which is crucial for a correct involution of the mammary gland. Both photoperiod and heat induce increased production of prolactin and, paradoxically, a lower expression of the hormone receptors in mammary tissues [43], which underlies a lower division of mammary secreting cells.

Compared to cows cooled during the whole dry period in summer, those without cooling show 4 to 5 kg/day less milk production during the whole subsequent lactation [38]. Compared to cows cooled during the dry period, uncooled cows have a lower proliferation of mammary epithelial cells 20 days before expected calving [44], but a similar mammary gene expression of proteins related to the synthesis of milk components in the following lactation [38]. These data suggest that heat stress during the entire dry period does not influence the synthetic capacity of epithelial cells during the following lactation, but it may affect udder growth.

## 6. Overcrowding in the Dry Cow Box and Animal Welfare

Concerning farm management, the resumption of calvings in late summer and early autumn causes an overcrowding in the group of dry cows in the middle of summer. The density of the animals is closely related to the space available for access to the feeder. As a result of the increase in stock density in the prepartum groups, food intake is reduced, metabolic problems increase and there is a high incidence of abomasum displacement after calving [45]. In a study completed in 2007, Huzzey et al. [46] recorded the feeding behavior during the last stage of gestation and observed that the cows developing post calving metritis had previously shown a decrease in the time of access to the trough, thus reducing dry matter intake 1–2 weeks before the parturition. The data indicated that social and eating behaviors affected the health and performance of cows in the transition period.

## 7. Overall Conceptual Framework

In the previous sections, we have put forward the multiple factors that underlie the systemic inflammatory state of dairy cows. With this background, we set out to identify the elements that could account for the long-term persistence of the negative effects of summer heat on autumn production and subsequent lactations.

Many studies have shown that the cells of the innate immune system display a “memory”, although different from that of the adaptive immune system. In essence, following stimulation by infectious and non-infectious stressors, the cells of the innate immune system would be able to build a state known as “trained immunity”, meant as outright “education” of the innate immune system: it consists of an epigenetic reprogramming, based on a change in transcriptional activity without permanent genetic changes. These changes in functional programming are prolonged over time and make these cells more reactive to a second stimulation [14]. On the whole, there is strong evidence of major epigenetic regulations of innate immunity genes in ruminants [47,48], which is also viewed in the prospect of a new disease control strategy [49,50,51]. Such epigenetic changes could take place also outside the innate immune system, as previously shown in reproductive biology models [52]. Owing to the above, our hypothesis is that the innate immune system in the dry and transition periods is repeatedly activated by infectious and non-infectious stressors, which is likely to be exacerbated by summer heat stress. The main features of “trained immunity” that could underlie the decrease of milk production in autumn are persistence, damaging effects and the metabolic shift it leads to.

### 7.1. Persistence

The long-term effects of “trained immunity” support the hypothesis that the combination of stressful events that occur in summer may have repercussions in autumn and in subsequent lactations, too; this implies a persistent reprogramming of the response to the initial stimulus in the host.

In vitro studies indicate that even after removing the initial stimuli, inflammatory phenotypes remain present in monocytes and macrophages. The long-lasting effects of “trained immunity” can be accounted for by epigenetic reprogramming in affected myeloid cells [14].

### 7.2. Damaging Effects of Trained Immunity

The detrimental effects of trained immunity are due to the fact that there is no timely resolution of the inflammatory response to prevent damage to host tissues [53], which outlines a risk condition for subjects who develop a long-lasting inflammatory response maintained by the innate immune system.

Moreover, according to the “danger model” [54], the immune reaction is triggered and/or amplified by tissue damage, following the release of specific alarm signals (DAMPs) [11]. Accordingly, a long-lasting proinflammatory microenvironment can be created in the cow after exposure to metabolic stress and related tissue damage [11].

### 7.3. Metabolic Shift

Metabolic shift is another feature of trained immunity. In fact, the epigenetic reprogramming at the basis of “trained immunity” causes a change in the metabolism of immune cells from aerobic oxidative phosphorylation to anaerobic glycolysis, in order to increase their reactivity to a possible second stress stimulation, but at the price of a higher consumption of glucose [55]. Hence, there is a continuous competition for glucose between the immune system and the mammary gland, where this molecule is necessary as a precursor of lactose and osmotic regulator of milk volume [56]. Immunoactivation markedly disrupts glucose homeostasis and may result in hypoglycemia and hyperlactemia [57,58,59].

In vitro experiments have shown a substantial increase in glucose consumption by activated immune cells, as glucose is their primary fuel and an important biosynthetic precursor [60,61].

An in vivo study by Kvidera et al. [62] demonstrated that acute endotoxemia induces hypoglycemia following an increased glucose utilization by the immune system. To ensure that the immune system is further adequately fueled, glucose production in the liver increases through both glycogenolysis and gluconeogenesis [57,58,63]. Peripheral insulin resistance occurs synchronously, leading to a decrease in glucose uptake by skeletal muscle and adipose tissue [64,65]. Decreased milk synthesis is one of the first observable signs of infection or inflammation in dairy cattle, and this presumably represents a strategy to conserve glucose and make it available to the immune system [62]. The reader is referred to a recent review which covers the issues of glucose demands, inflammation, metabolic adaptation and milk synthesis in dairy cattle [66]. On the whole, the innate immune system exerts its own functions under conditions of serious competition for energy with the mammary gland, whose capacity is likely to be down-regulated following the non-lactating summer period.

## 8. Why Does “Trained Immunity” Provide a Better Explanation of the Low Milk Yield Syndrome?

A few studies highlighted histological damage of the bovine mammary gland in dry cows exposed to summer heat stress [67]. Also, in vitro data [68] confirmed a negative impact of the heat stress, causing greater rates of programmed cell death in primary cultures of mammary gland epithelial cells. On the basis of these findings, one might wonder why be concerned about the innate immune system, and why advocate its important role in the autumn syndrome. As a matter of fact, the evidence confirming an important role of the innate immune system is diverse and substantial.

First, circumstantial evidence shows that the syndrome affects the whole herd, i.e., also the cows lactating in summer; these cows, after the physiological drop of milk yield in the hot period, do not show later on the expected yield observed in spring at similar DIM levels. Therefore, both dry and lactating dairy cows contribute to the observed syndrome, which cannot be solely accounted for by disturbances in the involuting mammary gland.

Interestingly, indigenous dairy cattle (Reggiana breed) with no summer anestrus do not show the autumn syndrome (Figure 3). This implies that a more regular distribution of calvings over the year, a better profile of innate immune responses [69] and the lack of a substantial NEB concur to a satisfactory milk yield of indigenous cows in autumn.

The liver of dry cows presents long-lasting functional problems, well beyond the hot summer period [42].

The effects of summer heat result in an ever-lasting reduction of milk yield observed in the progeny, despite the absence of an involuting mammary gland during heat stress [39]. The histological data about the direct impact of summer heat stress are contradictory. Compared with cooled cows, the number of alveoli would be reduced but the total secreting area would be unaffected [67]; the prevalence of apoptosis during udder involution would not be different [44], and the integrity of epithelial cells would be unaffected, as well [68]. On the whole, these findings point at reduced function of mammary alveoli, rather than direct damage, as a foundation of low milk yield after calving.

Most important, low milk yield in autumn goes along with increased prevalence of clinical mastitis. After a peak of clinical mastitis cases in July due to accumulated heat load (AHL), the prevalence starts to rise again in October and there is a second, distinct peak from November to January [70]. Therefore, the same mechanisms of dysregulated, inflammatory and metabolic memory response can be also conducive to a larger prevalence of clinical mastitis, after heat stress has subsided.

Finally, in the heat stress model on lactating dairy cows, low milk yield is only partly accounted for by reduced dry matter and energy intake [71], which implies worse energy balance and efficiency as a result of an inflammatory response [17]. In this respect, epigenetic changes in myeloid cells could underlie a temporal extension of the inflammatory response and related metabolic shifts [55].

Owing to the above, we can put forward the following points:(1)The autumn syndrome cannot be entirely accounted for by direct damages on the involuting mammary gland in summer.(2)A mechanism of temporal expansion of the primary stress is needed to provide a credible explanation of the observed phenomena. In this respect, epigenetic regulation of innate immunity genes (DNA methylation, histone modifications, nucleosome remodelling, non-coding RNAs, RNA modifications) [72] can be a robust explanation of the long-lasting effects of summer heat conditions. These could be investigated on proper tissue samples collected from the same cows before dry-off and after calving, respectively, by comparative analyses of e.g., chromatin compaction around innate immunity genes [48] and histon-modifying enzymes [47]. Also, the inclusion of cooled and control cows in the same herd would allow for a better focus of such cohort studies.(3)Accordingly, “trained immunity” is a robust conceptual framework to compose the above puzzle, which might be sided by analogous epigenetic changes on mammary gland precursors in the fetal period [39].(4)The “memory” of past stressors and the persistent inflammatory response defines the scenario in which the low milk yield autumn syndrome should be reasonably set and interpreted.

The aforementioned epigenetic changes (point 2) may take place following exposure to both infectious and non-infectious stressors; yet, we do not imply that such variations in gene expression are permanent and transferred to a subsequent generation.

## 9. Intervention Strategies

The scope of the control actions to alleviate the decrease of milk production in autumn is potentially wide. The objective is to reduce as much as possible the exposure of cows to summer heat stress, which implies in turn a substantial improvement of breeding structures, management, nutrition, and animal welfare at large. This way, the large-scale use of water soakers and fans in summer along with shaded barns causes significantly higher production levels [67]. This approach should be part of more ethical husbandry of intensively-managed dairy cows, which could overcome the current figures about life expectancy and fertility [4], as well as the ethical concerns for the present dairy farming practices [73].

Secondly, having in mind the described noxious effects of summer heat stress on dry cows, one might wonder if insemination services could be concentrated in spring, before the summer anestrus of Friesian cattle, in order to get the bulk of calvings in the winter period. This way, the crucial transition period of most dairy cows would not take place in summer.

Thirdly, there is substantial evidence of a negative impact of abrupt dry-off, with negative repercussions on well-being and metabolic response; this is further worsened in the presence of high residual milk production (>15 kg·d^−1^) [74]; interestingly, high milk production levels at dry-off also underlie increased probability of intramammary infections at calving [75]. This makes a case for a stepwise reduction of milk yield by reduced energy in the diet over 7–10 days before dry-off. This is a point of utmost importance, since the inflammatory response in the dry period has a serious impact on animal health in the following lactation [16].

## 10. Conclusions

The conclusions of this review paper are summarized in our SWOT Table (Table 1) and a causal pathway is shown in Figure 4 according to our working hypothesis.

Whereas circumstantial evidence, field remedies, evidence of epigenetic changes in other disease models of cattle [48] and the time scale of the event are in line with our conceptual framework, a formal demonstration of our hypothesis is still lacking, and we are perfectly aware of the technical constraints underlying a proper experimental plan to investigate this issue. Yet, we presume to insist on the opportunities provided by our hypothesis, which may allow for a better insight into adaptation strategies of dairy cattle and their production fitness. On the whole, we cannot but confirm the potential of “trained immunity” as a largely unchartered path to a better comprehension of fundamental biological phenomena with useful translational fallouts. The recent debate about the use of “trained immunity” for the control of the present COVID-19 pandemics is a most cogent demonstration of this tenet (https://www.quantamagazine.org/trained-immunity-offers-hope-in-fight-against-coronavirus-20200914/). As for animal science, “trained immunity” could be properly exploited towards better prophylaxis schemes of current production diseases, in which the memory of past stressors could play an important role in the occurrence of clinically overt disease and productive disorders.

## Figures and Tables

**Figure 1 animals-11-00388-f001:**
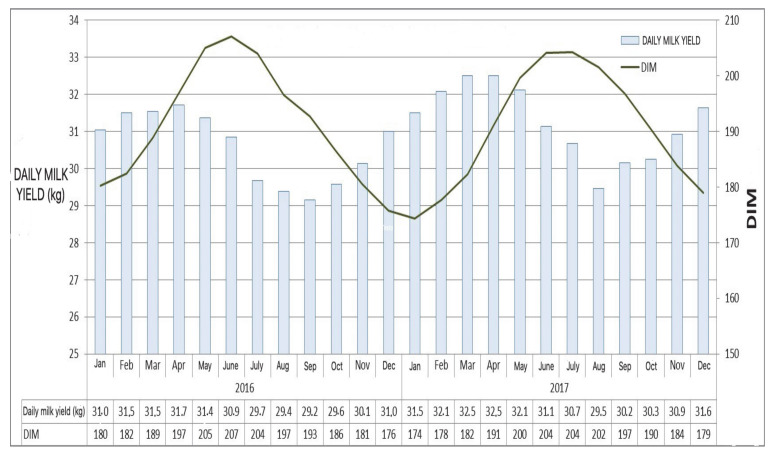
Average daily milk yield (kg) of Italian Holstein Friesian cattle in 2016–2017 as a function of Days in Milk (DIM). Data kindly provided by Alessia Tondo (AIA Studies Office, Italy). Compare spring (March–April) with autumn (October–November) milk yield data.

**Figure 2 animals-11-00388-f002:**
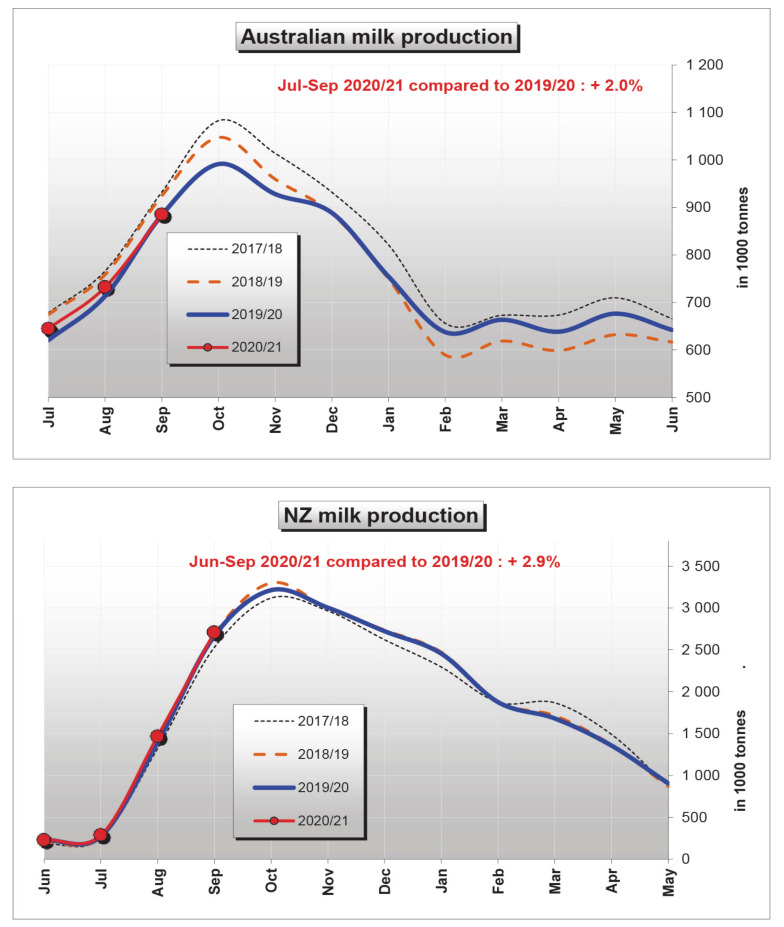
National milk production (tonnes × 10^3^) over four years in Australia and New Zealand (NZ). See: https://ec.europa.eu/—update 10 December 2020. Data source: Dairy Australia and Dairy Companies Association of New Zealand (DCANZ).

**Figure 3 animals-11-00388-f003:**
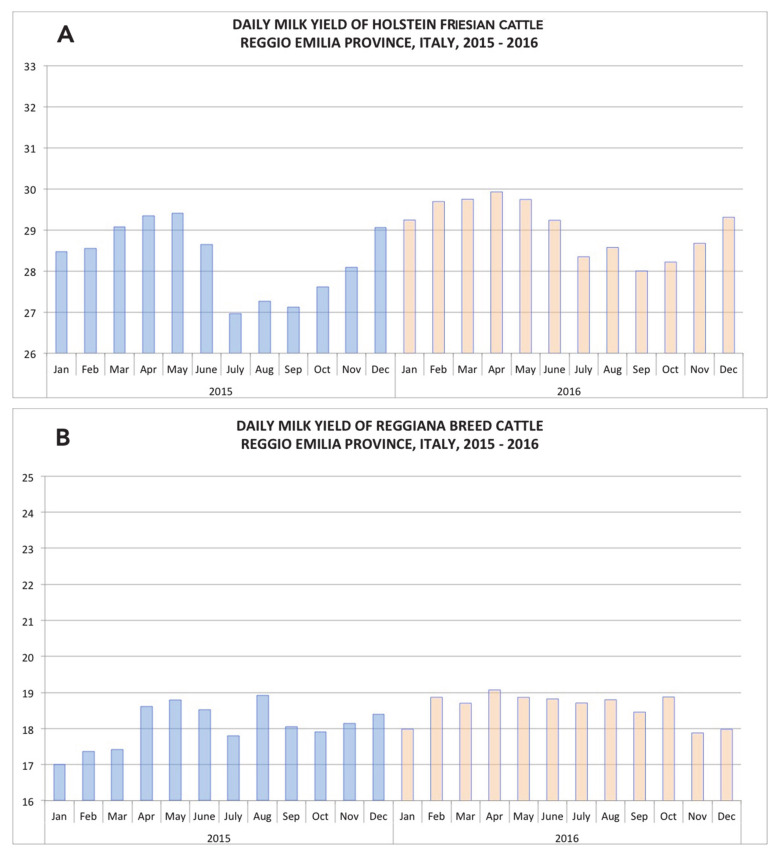
Comparison between average daily milk yield (kg) on a monthly basis of Holstein Friesian (panel (**A**)) and indigenous Reggiana dairy cattle (panel (**B**)) in Reggio Emilia Province, Italy. Period: 2015–2016. Courtesy of Alessia Tondo, AIA Studies Office, Italy.

**Figure 4 animals-11-00388-f004:**
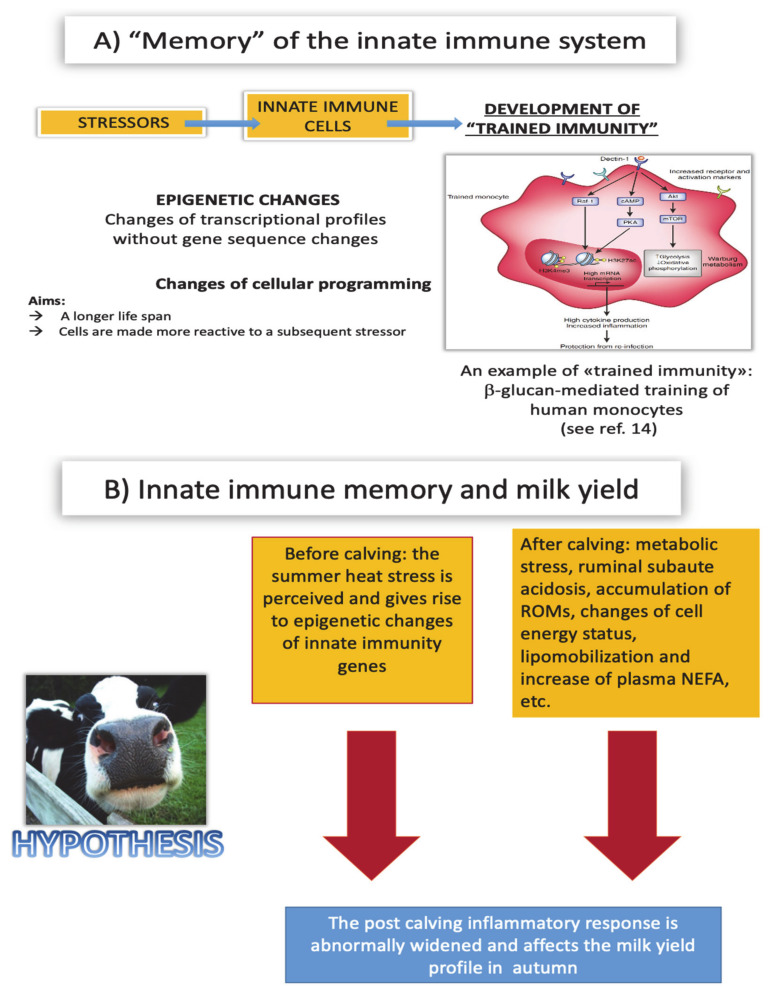
Panel (**A**) depicts the foundation and aims of innate immune memory. Panel (**B**) illustrates the possible causal pathway liaising summer heat stress, innate immune memory and milk yield profile in autumn. ROMs: Reactive Oxygen Metabolites. NEFA: non-esterified fatty acids.

**Table 1 animals-11-00388-t001:** The innate immune memory response to non-infectious stressors underlies the autumn low milk yield syndrome: robustness of the conceptual framework. SWOT table.

STRENGTH	WEAKNESS
Multi-disciplinary, comprehensive approach, combining field and experimental evidence.	Conceptual complexity of the experimental approach.
Clear time connection between event and previous exposure to summer heat.	Need for integration of widely different expertise sectors.
The syndrome is not seen in autochtonous cattle breeds with no summer anestrus and more regular distribution of calvings all over the year.	Lack of a validated trained immunity model in cattle.
Previous evidence of a major impact of stressors in the dry period on milk yield.	No studies about epigenetic changes in innate immunity genes of affected cows.
Previous evidence of life-long reduced milk yield in Friesian cows born from dams with summer dry period.	Overlapping of diverse stressors may be a serious confounding element.
Evidence in cattle of epigenetic changes of genes involved in innate immunity.	A cause/effect relationship can be hardly defined.
Reduction of heat stress in summer causes lesser drop of milk yield in autumn.	Experimental studies present utmost complexity and several operational constraints.
**OPPORTUNITIES**	**THREATS**
Establishment of useful investigation models into the crucial issues of adaptation physiology and sterile inflammation.	Continuous evolution of the dairy farming sector.
A wide scope for translation of the main scientific findings into working protocols in the animal health and welfare sectors.	Poor awareness of the real costs incurred as a result of the autumn low yield syndrome.
Establishment of new strategies for better profitability of dairy farming activities.	Poor awareness of the role of innate immunity in the response to non-infectious stressors.
	Poor awareness of long-term impact of heat stress
Establishment of adequate investigation tools for an effective monitoring of cattle herds.	Poor awareness of the crucial links between heat stress and quality of primary production.
A modern concept of the requirements to be met by welfare-friendly farms.	Farmers and veterinary practitioners often think of the syndrome as an unavoidable cost of farming activities.
	Poor awareness of the crucial links between heat stress and quality of primary production.

## Data Availability

This is a study solely based on publicly available scientific papers and reports of governmental and professional organizations. The reader is referred to the list of references and the citations inserted into captions.

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
