# Peer review of "The Autumn Low Milk Yield Syndrome in High Genetic Merit Dairy Cattle: The Possible Role of a Dysregulated Innate Immune Response"

_animals, 2021, doi:10.3390/ani11020388_

Round 1
Reviewer 1 Report
Please find the comments attached.

Author Response
The reply to reviewer 1 is uploaded as Word file.

Reviewer 2 Report
This manuscript proposes a conceptual framework to suggest that lower milk production in autumn is due to immune system responses when cows experience heat stress during the dry period in summer. This is an interesting concept that would make a novel contribution to the literature. The manuscript is generally clear but requires some editing of English language. My main reservation concerns evidence to support the proposed framework. Authors state that experimentation to test their framework would be challenging, but they could strengthen their argument considerably with evidence already available (assuming evidence is robust enough). I am not saying that their framework is invalid, just that they do not provide sufficient evidence in the manuscript.
Authors present findings from studies in a range of diverse areas to support their hypothesis. However, there is a general lack of numerical evidence by which to judge the contribution or importance of the different factors. Throughout the manuscript authors make statements followed by a reference number. The reader must take what authors say at face value or else read each reference to check that authors are quoting results that really support their argument. For important points, authors should briefly state the circumstances and results of other studies. It is insufficient to say that A is greater than B [ref]. What were the actual levels for A and B? Was the difference significant? What were the size and conditions of the study? Were there any caveats that limit interpretation of results in a general context? In other words, critical review is required.
Another concern is that authors focus on their immune-system framework and largely ignore major influences, such as nutrition, that might also explain autumn low yield syndrome. For example, when they describe the syndrome in 138-144, is this not due to difference in diet quality? Spring grass has higher energy and nutrient contents, and is more digestible, than typical winter diets. At any time of year, if you feed cows on higher quality diets, they will consume more nutrients and produce more milk. Even without a change in feed intake, if you decrease metabolisable energy concentration of a diet by just 0.6-0.7 MJ/kg DM, a typical 40 kg cow would decrease milk yield by 2.5-2.7 kg/d (the difference quoted in line 143). Authors should provide evidence to counter such basic arguments.
Other points to address:
145 Quintals? There are several definitions. Please use SI units or tonnes.
148 Data from the southern hemisphere (Figure 2) are not so convincing as Italy (Figure 1), especially New Zealand where cows are predominantly spring calving. Figure 2 does not show days in milk, so we cannot tell if cows experience heat stress during their dry period. I suspect the majority do not because spring is September to November. I suggest Figure 2 is deleted unless more evidence can be provided.
326 Glucose is not an element – it is a molecule.
353 Not clear what is ‘the latter’ – is it ‘cows lactating in summer’?
356 Autochtonous (autochthonous) – indigenous is a more common word.
367 Mammary alveoli
Table 1 – spelling of STRENGTH
Author Response
The reply to reviewer 2 is uploaded as a Word file.
Round 2
Reviewer 1 Report
I will email review independently as unable to relocated from a word file into this platform.

Author Response
The reply to reviewer 1 can be found in the attached file.

Reviewer 2 Report
The authors misunderstood my comment about evidence. I did not mean there was no evidence, I meant that evidence could be presented more strongly. My suggestion was to describe studies better, rather than just making statements with a reference number after them. However, additional material inserted, plus badging the manuscript as an opinion paper, largely overcome my concern.
My other comments have been addressed, although some spelling mistakes remain.
Author Response
The reply to reviewer 2 can be found in the attached file.
